# Electronic Nose and Head Space GC–IMS Provide Insights into the Dynamic Changes and Regularity of Volatile Compounds in Zangju (*Citrus reticulata* cv. Manau Gan) Peel at Different Maturation Stages

**DOI:** 10.3390/molecules28145326

**Published:** 2023-07-11

**Authors:** Peng Wang, Haifan Wang, Jialiang Zou, Lin Chen, Hongping Chen, Yuan Hu, Fu Wang, Youping Liu

**Affiliations:** Department of Pharmacy, Chengdu University of Traditional Chinese Medicine, State Key Laboratory of Southwestern Chinese Medicine Resources, Chengdu 611137, China; wangdapeng19981026@163.com (P.W.); haifan.wang@foxmail.com (H.W.); wangfuboy2019@126.com (F.W.)

**Keywords:** zangju peel, electronic nose, headspace-gas chromatography–ion mobility spectrometry, partial least squares-discriminant analysis, variable importance in the projection

## Abstract

Zangju (*Citrus reticulata* cv. Manau Gan) is the main citrus cultivar in Derong County, China, with unique aroma and flavour characteristics, but the use of Zangju peel (CRZP) is limited due to a lack of research on its peel. In this study, electronic nose, headspace-gas chromatography–ion mobility spectrometry (HS-GC–IMS), and partial least squares-discriminant analysis (PLS-DA) methods were used to rapidly and comprehensively evaluate the volatile compounds of dried CRZP and to analyse the role of dynamic changes at different maturation stages. The results showed that seventy-eight volatile compounds, mainly aldehydes (25.27%) and monoterpenes (55.88%), were found in the samples at four maturity stages. The contents of alcohols and aldehydes that produce unripe fruit aromas are relatively high in the immature stage (October to November), while the contents of monoterpenoids, ketones and esters in ripe fruit aromas are relatively high in the full ripening stage (January to February). The PLS-DA model results showed that the samples collected at different maturity stages could be effectively discriminated. The VIP method identified 12 key volatile compounds that could be used as flavour markers for CRZP samples collected at different maturity stages. Specifically, the relative volatile organic compounds (VOCs) content of CRZP harvested in October is the highest. This study provides a basis for a comprehensive understanding of the flavour characteristics of CRZP in the ripening process, the application of CRZP as a byproduct in industrial production (food, cosmetics, flavour and fragrance), and a reference for similar research on other *C. reticulata* varieties.

## 1. Introduction

Citrus is grown in more than 140 countries and territories worldwide and mainly includes mandarin (*Citrus reticulata*), grapefruit (*Citrus maxima*), citron (*Citrus medica*), oranges (*Citrus sinensis*) and limes (*Citrus aurantium*) [1]. In particular, *C. reticulata* of local cultivars is very popular, such as sweet orange (*C. sinensis*), *C. unshiu*, *C. reticulata* ‘Chazhi’, *C. reticulata* ‘Ponkan’ and Zangju (*C. reticulata* cv. Manau Gan), among others. Zangju is known locally (in the Tibetan region) as ‘Jiaxu’, which means ‘fruit of longevity’, which may be related to the fact that the Zangju tree has been growing for hundreds of years (the common citrus tree lives between 40 and 50 years). Zanju is the local cultivar of *C. reticulata* cv. Manau Gan and is popular with people because of its unique aroma and flavour and easy peeling. It is mainly distributed in Derong, Muli and other counties [2] in the agricultural and pastoral areas of the middle and lower reaches of the Sanjiang River Valley (subtropical valley climate) in the southern section of the Hengduan Mountains, China. It has been cultivated for more than 400 years, with a planting area of over 11,600 hectares and an annual output of approximately 130,000 tons [3]. At present, the fruit is mainly used for various purposes, such as directly eating the pulp, processing it into juice, preserving fruit, cooking seasoning, etc., and using the *C. reticulata* cv. Manau Gan peel (Zangju peel, CRZP) to relieve headaches and coughs. Local residents believe that it is a kind of citrus with the homology of medicine and food, with good economic value and medicinal potential. However, there is less use of the CRZP, resulting in a waste of resources.

Citrus pericarp is a byproduct of citrus processing that is rich in volatile oil, flavonoids, alkaloids and other active ingredients and is one of the main sources of essential oil (0.5–3.0 kg/t citrus fruits) at present [4,5,6,7]. Previous studies have shown that citrus essential oils, including terpenoids, esters, alcohols and aldehydes, among others [8,9], have antioxidant, anti-inflammatory, antibacterial, anticancer, sedative, hypnotic and fatigue-relieving effects [10,11,12,13,14] and have been widely used in food, medicine, cosmetics, spices and other fields [11,15]. As fresh citrus peel is susceptible to microbial influences that can lead to contamination, it needs to be dried before being used mainly in cooking, food additives, and cosmetic additives. The dried peel is a traditional Chinese medicine used to treat coughs, indigestions and other diseases [16,17]. At present, there are three main drying methods for citrus peel, including natural drying, hot air drying and freeze drying, and different drying methods that influence the volatile compounds of citrus peel. For example, the dried peel of *C. reticulata* ‘Chazhi’ is mainly prepared by natural drying, while the dry peel of other citrus varieties is mainly prepared by hot air drying due to regional climate differences. Low-temperature hot air drying is conducive to the retention of VOCs in the drying process of citrus peel [18,19]. Therefore, this study comprehensively considered citrus varieties, VOCs, regional climate and preparation costs and used low-temperature hot air drying to prepare dried CRZP.

In addition, citrus varieties and different ripening stages also lead to changes in the types and contents of citrus essential oils [9,13,20,21]. Currently, most studies focus on the changes in nonvolatile compounds and biomarkers of citrus, such as *C. reticulata* ‘Chachi’ and *C. wilsonii*; however, there are few studies on the changes in volatile compounds at different maturation stages. Hou et al. studied the changes in VOCs in navel oranges at different growth stages and found that 55 volatile compounds showed significant differences during the maturation process. The relative concentrations of terpenoids and aldehydes increased with increasing maturity, reaching their maximum in November, while esters showed the opposite trend [21]. Kim et al. investigated the volatile compounds of satsuma mandarin at different maturity stages and found that the linalool content increased by approximately three times during maturity, and the sesquiterpene content showed a trend of first increasing and then decreasing [22], which was different from the results of Hou et al. Obviously, there are differences in the content of VOCs in citrus peel at different stages. By studying the content changes of compounds to determine the maturity of citrus, it is convenient to collect and improve the content of active compounds in citrus. However, due to the lack of research on CRZP, the composition and content of essential oils are unknown, especially in different maturation stages, leading to the limited application of CRZP.

In this study, we collected Zangju from five harvesting periods (October 2022 to February 2023, sampled once a month) and divided them into four mature stages: immature stage (October), incomplete mature stage (November), commodity mature stage (December), and complete mature stage (January and February the following year), using hot air drying to prepare the CRZP. Electronic nose (E-nose) and headspace-gas chromatography–ion mobility spectrometry (GC–IMS) were used to detect the VOCs at different maturation stages; linear discriminate analysis (LDA) and principal component analysis (PCA) were used to evaluate the detection and identification effects of the two methods; and the fingerprint of VOCs of CRZP at different picking stages was determined. The PLS-DA model was established to identify the key aroma compounds in different harvesting periods using VIP values as evaluation criteria. The purpose of this study is to reveal the aroma characteristics and various rules of VOCs in CRZP at different ripening stages, providing a theoretical basis for the development and utilization of the VOCs in CRZP. For other *C. reticulata* varieties, research on the volatile compounds of these varieties provides a reference.

## 2. Results and Discussion

### 2.1. Results of E-Nose

LDA can minimize intragroup variance and maximize intergroup variance; in other words, it can reduce intragroup differences and expand intergroup differences [22,23]. Therefore, LDA was used to distinguish CRZP in the five picking stages. As shown in Figure 1A, the variance contribution rates of LD1 and LD2 are 82.58% and 13.52%, respectively. Five groups of samples are far apart, especially the distance between ZJP and the other four groups, indicating that ZJP is significantly different from the other samples.

Figure 1B shows that the W5C, W1S and W1W sensors have stronger induction of VOCs in the samples, indicating that CRZP may contain higher levels of alkane aromatics, terpenes and sulfur-containing organic compounds. However, E-nose cannot quantitatively or qualitatively identify the important aroma substances in citrus peel, such as alcohols, aldehydes and esters [9,18,24]. This indicates that the E-nose is not completely sensitive to CRZP aroma substances and can only represent part of the aroma characteristics. The results showed that there were significant differences in CRZP at different harvesting stages, which could be further identified and compared.

### 2.2. HS-GC–IMS Visual Topographic Plot Comparison

In this study, HS-GC–IMS was used to analyse the volatile compounds in dried CRZP at different maturation stages to help identify the volatile compounds and their changes during the ripening process. To observe the differences in different periods more directly, FlavourSpec^®^ (Version 2.000) instrument software was used for visual analysis of the measured results, including three-dimensional spectra and corresponding two-dimensional spectra. The combination of the two spectra can more clearly reflect the differences and changes in volatile substances [24,25].

The *x*-, *y*- and *z*-axes of the three-dimensional spectrum represent the migration time, retention time and peak intensity, respectively. The differences in VOCs in different samples can be visually seen in the three-dimensional diagram (Figure 2A). The results show that the peak signal distribution of volatile compounds in CRZP at different stages is basically similar, but the peak signal intensity is different. This shows the variation in volatile organic compound content in different samples. Compared with the three-dimensional spectrogram, the two-dimensional spectrogram has the same information except for the lack of a *Z*-axis, and the two-dimensional spectrogram can directly show the difference in compounds and concentrations among different samples. The background of the figure is blue (Figure 2C), and the red vertical line at abscissa 1.0 is the RIP peak (reactive ion peak after normalization). Each dot in the diagram represents a volatile substance or dimer, and the colour represents the concentration of the substance, with white representing a lower concentration, red representing a higher concentration, and a darker colour representing a higher concentration.

To better compare the differences among samples, combined with the results of the E-nose, the spectrograph of samples at the commodity maturity stage in December was selected as a reference to obtain two-dimensional top views of samples at other maturity stages (Figure 2D). White represents similar or consistent with the reference value; red represents higher than the reference value; and blue represents lower than the reference value. As shown in the figure, the composition and content of volatile compounds in tangerine peel changed significantly at different ripening stages. The results showed that the content of volatile compounds in immature tangerine peel changed significantly with increasing maturity, and the number of red dots was higher than that of blue dots in the full maturity period (January to February of the next year), indicating that the content of most volatile compounds increased after full maturity. Similar reports have also been reported in studies on fruits such as camellia tangerine, green plum and *Passiflora edulis* [24,26,27], which are related to the vigorous metabolism of ripe fruits.

### 2.3. Identification of Substances

HS-GC–IMS was used to qualitatively analyse the VOCs from CRZP at different maturation stages. Ninety signal peaks were determined by comparing the drift time and retention index with the control, and 78 volatile compounds were preliminarily identified, as shown in Table 1. It mainly includes ten alcohols, twenty-three aldehydes, nineteen monoterpenoids, nine ketones, eight esters, and nine other volatile substances, with relative contents of 2.13%, 25.27%, 55.88%, 4.11%, 3.87% and 8.74%, respectively (Figure 2B). In particular, 18 substances, including α-terpineol, citronellal, linalool, decanal, cyclohexanone and ethyl acetate, formed dimers, which were mainly related to the concentration of volatile compounds and proton affinity. The protons of volatile substances with higher proton affinity than water and higher concentrations can be transferred to substances with high proton affinity, thus contributing to the formation of dimers [24,28].

#### 2.3.1. Alcohols

A total of ten volatile alcohol compounds were identified in five different harvesting stages of CRZP, the number of which was third only to aldehydes and monoterpenoids. Examples include 1-penten-3-ol, (z)-2-penten-1-ol, and tetrahydrolinalool. Studies have shown that linalool and tetrahydrolinalool are important volatile compounds in citrus peel and have the aroma of citrus fruits, which are usually produced through the MEP pathway [24,29]. Other alcohols, such as α-terpineol and benzyl alcohol, were also considered to have the aroma characteristics of jasmine and lilac, and benzyl alcohol was a unique component of CRZP compared with *C. reticulata* ‘Chachi’, *C. reticulata* ‘ponkan’ and navel orange. In addition, α-terpineol, linalool and tetrahydrohydrolinalool have antibacterial, antioxidant and sedative effects [30]. Benzyl alcohol has anti-louse, anti-insect, analgesic and antiseptic properties. These include being used for Benzyl Alcohol Lotion 5% [31], anti-lice treatments, and injection preparation solvents, such as the solvent of spectinomycin hydrochloride that is used for injections [16].

#### 2.3.2. Aldehydes

A total of 23 volatile compounds of aldehydes were identified in the CRZP, most of which were small molecules of C4–C9. Previous studies have shown that these small-molecule aldehydes, which are widely found in the fruits and peels of citrus plants such as Shatian pomelos, oranges, lemons and tangerines, are highly volatile, have a low threshold, and are a major component of fresh and unripe fruit aromas. They are usually produced by the degradation of fatty acids [4,27,32,33]. Dodecanal, Decanal, etc. are C10-C12 molecules with high volatility and a citrus aroma. Among them, dodecanal and citronellal also have the aromas of pine leaf oil and cherry, respectively. These three compounds exist in the essential oils of lemon, citrus and sweet orange peel and mainly serve as spices and edible flavours [5,34,35]. Other volatile compounds, such as hexanal, heptanal, octanal, etc., have a fruity aroma, and benzaldehyde has a bitter almond taste, which is often used in the production of spices. Heptanal is also the raw material for pharmaceutical and rubber products. In addition, isovaleraldehyde, butanal and other aldehydes are intermediates that produce fragrances, spices, and other organic compounds [5,9,18]. Interestingly, the citronellal identified from CRZP in this study was not found in previous studies of other cultivated species of *C. reticulata* (*C. reticulata* ‘Chachi’, *C. unshiu* and *C. reticulata* ‘Ponkan’, etc.) [21,27,36].

#### 2.3.3. Monoterpenoids

A total of 19 monoterpenoids were identified. α-Terpinene, α-pinene, β-pinene, terpinolene, β-myrcene, (E)-ocimene, α-thujene, norbornane, 7,7-dimethyl-2-meth, γ-terpinene, 3-carene, α-phellandrene, etc., and sesquiterpene (β-caryophyllene) are also found in *C. reticulata* ‘Chachi’ [27]. Most of these terpenoids are produced by the mevalonate pathway (MEP) [5]. These C10 monoterpenoids accumulate in citrus peel and give the typical aroma of citrus, sweet orange and lemon. They are the main source of citrus aroma and mainly serve as intermediates between spices or synthetic flavours and spices. In this study, 3-carene, α-thujene, norbornane, and 7,7-dimethyl-2-meth were found in CRZP, which was not reported in previous studies with Citrus (*C. reticulata* ‘Chachi’, navel orange). It is worth noting that 3-carene has a strong pine-like aroma, which may be related to genetic factors and the unique growing environment of Tibetan orange [9,27,37,38], and is often used as a raw material for the synthesis of food flavours and spices.

#### 2.3.4. Ketones, Esters and Others

A total of nine ketones were identified in CRZP. These compounds included cyclohexanone, 2-hexanone, 3-hydroxy-2-butanone, 2,3-butanedione, acetone, 1-penten-3-one, and 2-butanone. Most of these ketones are produced through β-oxidation and degradation of fatty acids [39]. The short-chain ketones of C3-C6, such as cyclohexanone, 2-hexanone, 3-hydroxy-2-butanone, 2,3-butanedione, 1-penten-3-one, etc., are considered to have rich fruit flavours in fruits and green leafy plants. The aromatic compounds of mint and milk are often used as raw materials for the production of citrus and dairy food flavours, solvents and spices [27,40].

The eight esters identified in this study included crotonic acid, hexyl ester, ethyl salicylate, 2-nonynoic acid methyl ester, ethyl acetate, butyl acetate, styralyl acetate, and methyl salicylate. Esters have fruit and wood aromas, which are an important component of citrus aroma and an important evaluation index of citrus aroma. It is mainly produced through the esterification of alcohols and acids, which has poor stability and is easy to decompose in the ripening process [9,41]. Crotonic acid, hexyl ester, ethyl salicylate, ethyl acetate, methyl salicylate, ethyl acetate, methyl salicylate, etc. [27,37]. Ethyl salicylate and methyl salicylate were reported for the first time in this study, which may be the reason for the distinctive aroma of Zangju.

Other volatile compounds found included cis-anethol, methyl eugenol, 4-allylanisole, allyl propyl disulfide, 2-methoxy-3-isobutyl pyrazine, 4-ethylphenol, 1-mercaptopropane and 2-methylpropanoic acid. Interestingly, the only organic acid found in the volatile compounds of tangerine is 2-methylpropanoic acid, which is mainly used for organic synthesis and does not characterize aroma. We also found several sulfur-containing compounds and nitrogen-containing compounds, which are the main odour compounds in litchi and have anti-insect effects [41]. In addition, methyl eugenol, 4-allylanisole, methoxy-3-isobutyl pyrazine, cis-anethol and other substances are often used as spices [4,27,42,43].

### 2.4. Analysis of VOC Fingerprints

In order to more intuitively show the variation rule and relative content comparison of volatile substances in the CRZP at different maturity stages, we used the Gallery Plot plug-in to draw the fingerprint of volatile substances. Compared with the topographic map showing the trend of volatile compounds, fingerprints can intuitively and quantitatively compare the changes in volatile compounds (increase, decrease, or fluctuation) between different samples through the strength of different spots and reveal the dynamic changes in each substance [27,44]. In the fingerprint, each row represents a sample (three parallel at each stage), and each column represents a volatile component (the darker the colour, the higher the content). The VOC fingerprint of the CRZP at five harvesting stages is shown in Figure 3. The fingerprint showed the diversity of volatile compounds in CRZP at different harvesting periods. A total of 90 peaks (78 VOCs were defined) were preliminarily identified. The seventy-eight different VOCs included ten alcohols, twenty-three aldehydes, nineteen monoterpenoids, nine ketones, eight esters and eight other substances.

VOCs in the CRZP mainly consist of alcohols, aldehydes and monoterpenes. For example, linalool M/D, α-terpineol M/D, decanal M/D, (E)-2-hexenal M/D, butanal M/D, citronellal M/D, α-terpinene M/D, α-pinene, β-pinene, terpinolene, α-thujene, γ-terpinene and other volatile substances are the basic compounds in the ripening process of Zangju. In addition, there are substances with high peak intensities in ZGP (October) samples, such as Crotonic acid, hexyl ester, Ethyl salicylate, Ethyl acetate M/D, Styralyl acetate, Methyl salicylate, Cis-Anethol, Methyl eugenol, 4-Allylanisole, 2-Methoxy-3-isobutyl pyrazine, 4-Ethylphenol, mainly esters and ethers. The characteristic substances present in the ZIP (November) sample are hexanal M/D and 3-methyl-2-butenal M/D. The ZJP (December) sample has three characteristic compounds: 2-hexanone, 3-hydroxy-2-butanone M/D, and 2-methylpropanoic acid. There were nine VOCs in the ZKP and ZLP samples (in the later period of fruit picking, from January to February of the next year). Including 2-Methyl-1-pentanol, 2-Methyl-1-butanol M/D, 3-Methyl-1-butanol, 1-Butanol, Cyclohexanone M/D, 2-Butanone, Crotonic acid, hexyl ester, 1-Mercaptopropane M/D.

### 2.5. Analysis of Dynamic Changes and Formation Regularity of VOCs

Further analysis of the fingerprint (Figure 3) and Table 2 showed that with increasing maturity, ketones first increased, reached the highest value in the commodity maturity stage (December), and then tended to be flat. Specifically, cyclohexanone M/D, 2-hexanone, 3-hydroxy-2-butanone M/D, and 2-butanone have a fruity aroma. During the ripening process, fatty acids accumulate in Zangju, providing sufficient substrate for the generation of ketones [45,46]. Alcohols, esters and other compounds showed a precipitous decline, then increased, and finally tended to flatten out. The lowest value was reached in the immature maturity stage (November), and the highest value was reached in the full maturity stage (January). It is manifested as α-terpineol M/D, tetrahydrohydrolinalool, ethanol, crotonic acid, hexyl ester, ethyl salicylate, 2-nonynoic acid methyl ester, ethyl acetate M/D, and styralyl acetate, with a citrus fruit aroma and wood aroma. This may be related to the influence of climatic conditions (temperature, precipitation, etc.) on enzyme activities related to the synthesis of such substances [27,47]. The aldehydes and terpenes showed a slow change trend of first decreasing and then increasing, reaching their lowest value in the full maturity stage (January). The specific pattern was dodecanal, octanal, benzaldehyde, (E)-2-pentenal M/D, pentanal, alpha-terpinene M/D, alpha-thujene, norbornane, 7,7-Dimethyl-2-meth, 3-Carene, α-Phellandrene, with citrus aromas of sweet orange, lemon, pine leaf oil and cherry. This may be due to the activation of the lip oxygenase (LOX) pathway with increasing maturity. Active hydroperoxide lyase (HPL) and alcohol dehydrogenase (ADH) react hydroperoxide with alcohol, thus reducing the alcohol content and producing other substances [24,26], which reaches the lowest value in the full maturity stage (January) and increases slightly in February. This may be related to the influence of the growth environment (temperature) on related enzyme activities [22,27,47]. The lower temperature inhibits enzyme activity, leading to a decrease in the biosynthesis of VOCs, resulting in a decrease in the relative content of VOCs in December and January. In general, with increasing maturity, the relative content of VOCs in tangerine peel showed a downward trend and then increased slightly in the CRZP harvested in February, indicating that the relative abundance of VOCs in the CRZP harvested in October was the highest compared with the other four periods.

To better analyse the dynamic changes in VOCs in the CRZP at different picking stages, we drew a clustering heatmap based on the fingerprint results (Figure 4). The darker the red or blue, the higher or lower the peak intensity, respectively. The samples are clustered into five different categories [24,48], and the samples with similar accumulation patterns have a higher correlation. The first category includes alcohols, aldehydes, ketones and monoterpenes, such as 1-butanol, citronellal M, 2-butanone, 3-methyl-1-butanol, 2-methylbutyraldehyde M/D, cyclohexanone D, 2-methyl-1-pentanol, α-thujene, β-myrcene, γ-terpinene, and octanal. These compounds produce citrus, cherry, and other fruit aromas whose intensity increases with maturity, reaching a maximum in January and then decreasing in February. Compounds 3-Methyl-1-butanol, α-thujene, and citronellal M are unique VOCs for samples taken in January. The second category contains alcohols, aldehydes, ketones, terpenoids, esters, 1-penten-3-one, (z)-2-penten-1-ol, α-terpinene M/D, crotonic acid, hexyl ester, 2-nonynoic acid methyl ester, ethyl salicylate, and dodecanal. The intensity of these C6-C10-producing VOCs with complex unripe aromas decreased with increasing maturity, which was related to the activation of the LOX pathway, resulting in a reduction in the biosynthesis ability of unripe VOCs, indicating that the samples collected in October had more complex unripe aroma characteristics. The third group includes ketones, monoterpenes, alcohols, and aldehydes, such as (E)-ocimene, β-pinene, α-terpinene, 2-methylpropanoic acid, 2-hexanone, 3-hydroxy-2-butanone, decanal, and linalool. The intensity of these fragrance-producing substances showed a trend of first increasing and then decreasing, reaching a maximum in December. The fourth kind includes most alcohols and aldehydes as well as some esters and terpenes, such as α-terpineol M/D, benzyl alcohol, tetrahydrohydrolinalool, (E)-2-hexenal M/D, heptanal M/D, (E)-2-pentenal M/D, and pentanal. The changing trend of styralyl acetate, ethyl acetate M/D, α-pinene, and α-phellandrene was similar to that of the second component. The fifth kind consisted of aldehydes and ketones, including 3-methyl-2-butenal M/D, hexanal, isovaleraldehyde M/D, butanal, and 2,3-butanedione, and their intensity showed an irregular pattern, increasing from October to November. Then it fell and rose again in January and February. This indicates that the samples collected at different maturity stages have unique VOCs, among which 3-methyl-2-butenal M/D is the characteristic VOC of the samples collected in November. In general, the results of cluster analysis are consistent with those of fingerprint analysis.

To clarify the characteristic differences among samples and the similarities and differences in VOC composition, PCA and PLS-DA models were adopted to further process the chromatographic peak data of CRZP at different maturity stages [24,27]. The VIP method was used to search for characteristic aromas. The total sample size of the model was fifteen (five periods × three). The Y variable in the model refers to the dried fruit skins collected in five harvesting periods (October 2022–February 2023). The X variable represents the 78 VOCs identified in this study.

Figure 5A shows the PCA scoring maps of VOCs of the CRZP at five different harvesting stages measured by GC–IMS. It can be seen from Figure 5A that after dimension reduction, the contribution rates of the first two principal compounds were 66.52%, and the first principal component (PC1) and the second principal component (PC2) were 40.73% and 25.79%, respectively. This shows that after feature compression, relatively complete information is still retained, and the feature difference of the original variable can be better represented. Samples ZGP, ZIP, ZJP, ZKP and ZLP were obviously separated for four different maturation stages, indicating that there were obvious characteristic differences among samples.

The PLS-DA results are shown in Figure 5B. The first component and the second component accounted for 47.6% and 27.6%, respectively, in the PLS-DA scoring chart. R2 and Q2 have high accuracy, both exceeding 0.91 and 0.6 (Figure 5D), indicating that the model has good accuracy and reliability [49]. The score map is used to directly visualize the similarities and differences between samples, and the remote clustering in the score map represents the differences between samples. Samples from five different harvesting periods can be separated well, indicating that the HS-GC–IMS method is very accurate in classifying the CRZP at different stages [24,26,47]. To further determine the key aroma compounds of CRZP in different harvesting periods, the VIP value was used to evaluate the degree of influence and explanatory power of each variable on sample classification. We took a VIP value greater than one as a significant criterion (Figure 5C) [24,27]. A total of twelve VIP > one volatile compound were identified, among which hexanal D and cisi-anethol were the key discriminant compounds in the CRZP collected in October. (E)-2-Hexenal D, norbornane, and 7,7-dimethyl-2-meth were the key discriminant compounds in the sample collected in November, while α-Pinene, Decanal M, Linalool M, and α-Thujene were identified in the sample collected in December. Octanal, 4-allylanisole, and β-myrcene were identified in the samples collected in January and February of the following year, and decanal D in the samples collected in February was the key volatile compound to distinguish the samples collected in January and February. These compounds can be used as potential markers to distinguish CRZP samples collected during different harvesting periods.

## 3. Materials and Methods

### 3.1. Plant Materials and Preparation

To exclude the influence of the growing environment and genetic factors on the experimental results, this study used the location and specific plants to study the peel of an orange at different maturity stages. All samples were taken from Bentu Township, Derong County, Garze Prefecture, Sichuan Province, China (99°16′37″ E; 28°32′32″ N; altitude: 2225.15 m); 4 adjacent Zangju trees with high yield and stable quality were randomly selected, numbered and listed. A total of 6 fresh fruits of similar size and free of pests and diseases were randomly picked from each tree, sampled once a month (October 2022 to February 2023), and numbered ZGP, ZIP, ZJP, ZKP, and ZLP in turn. According to the appearance and colour characteristics, the five harvesting periods were divided into four ripening stages: the immature stage (October) showed dark green peel, the incomplete maturity stage (November) showed chartreuse peel, the commercial ripening stage (December) showed aurantiacus pericarp and the fully matured stage showed orange–red pericarp (January and February). The collected samples were washed, peeled manually, and dried by hot air at a constant temperature of 50 °C for 18 h [20]. After preparation, all samples were stored in a dry place at room temperature. The CRZP samples for each stage are shown in Figure 6.

### 3.2. E-Nose Analysis

E-nose analysis was performed using an 8 MEMS-MOS E-nose (Isensortalk Co., Ltd., Beijing, China). First, 1.5 g of the sample was placed into a 20-millilitre headspace vial. Two probes were inserted into the headspace vial, one connected to the sensor chamber and the other to a charcoal filter. Each sample was run in triplicate, and data were obtained from the sensor array 60 s after injection to determine the stability of the sensor signals. Data from 55 s to 58 s were taken for subsequent statistical analysis. After sample analysis, the system was purified with filtered, clean air for 120 s to reestablish the instrument baseline. All samples were tested at room temperature.

### 3.3. HS-GC–IMS Analysis

In this research, VOCs were determined using a GC–IMS instrument (FlavourSpec^®^, G.A.S., Dortmund, Germany) equipped with an automatic sampler. First, 0.2 g of CRZP was accurately weighed, placed in a 20-millilitre headspace injection bottle and incubated for 15 min at 80 °C and 500 rpm. The volume of automatic headspace injection was 200 μL, and the temperature of the injection needle was 85 °C. Then, the extracted volatile compounds were pre-separated on a gas chromatographic column (MXT-5, 15 m, ID: 0.53 mm) and transported to IMS with carrier gas for a total of 40 min of analysis. Each sample was analysed in parallel 3 times. The column temperature and IMS temperature were 60 °C and 45 °C, respectively. The carrier gas/drift gas was high-purity nitrogen (99.99%). The flow rate of the drift gas was set at 150 mL/min. The flow rate of the carrier gas was 2 mL/min in the initial 2 min, increased to 10 mL/min in 2–10 min, increased to 100 mL/min in 10–20 min, and then maintained for 20 min. The whole process took 40 min. Volatile compounds were identified based on RI and the draft time of standard substances in the GC–IMS library (G.A.S.).

### 3.4. Statistical Analysis

The results listed in the table and figure are the average values of three experiments. VOCal software (v 2.2) views the qualitative and quantitative analysis of spectra and data and uses the built-in NIST and IMS databases of VOCal software to conduct qualitative analysis of substances. Each point obtained in the spectra represents a volatile component. The Reporter plug-in compares the spectral differences between samples (three-dimensional and differential spectrograms); the Gallery Plot plugin performs fingerprint comparison to visually and quantitatively compare the differences in volatile compounds between different samples. WinMester software (1.6.2.18, Airsense Analytics GmbH, Schwerin, Mecklenburg, Germany) was used to draw LDA diagrams and radar charts. PLS-DA analyses were performed using MetaboAnalyst 5.0. R software was used to visualize the PCA, cluster analysis and heatmap data.

## 4. Conclusions

In this study, E-nose and HS-GC–IMS were used to detect VOCs in five different harvesting periods of the CRZP. The VOC fingerprints were established based on the HS-GC–IMS and PLS-DA models. Seventy-eight volatile compounds were identified, including ten alcohols, twenty-five aldehydes, seventeen terpenoids, nine ketones, eight esters and nine other volatile substances. The E-nose is not sensitive to important aroma compounds such as alcohols and aldehydes in CRZP and can only represent part of the aroma, which includes alkane aromatics, terpenes and sulfur-containing organic compounds. The results of HS-GC–IMS showed that the contents of alcohols, aldehydes and monoterpenoids were relatively high. The VOCs in the CRZP increased, decreased and fluctuated during different harvesting periods. The alcohols and aldehydes that produce unripe fruit aromas, such as (z)-2-penten-1-ol, α-terpinene M/D and dodecanal, were relatively high in the immature stage (October to November). In contrast, monoterpenoids, ketones and esters with ripe fruit aromas, such as 2-butanone, 2-methylbutyraldehyde M/D, cyclohexanone D, α-thujene, β-myrcene, γ-terpinene, crotonic acid, hexyl ester, etc., are relatively high in the full maturity stage (January to February). The results of PCA and PLS-DA showed that the volatile organic compounds in the samples at different harvesting stages were well separated and easy to distinguish. A total of twelve key volatile compounds were identified by the VIP method to distinguish samples picked at different stages. In other words, the relative VOC content of CRZPs harvested in October was the highest. The results of this study can help us comprehensively understand the flavour characteristics of CRZP during the ripening process, provide a basis for the application of CRZP, a byproduct of Zangju, in industrial production (food, cosmetics, flavour and fragrance), and provide a reference for similar research on other *C. reticulata* varieties.

## Figures and Tables

**Figure 1 molecules-28-05326-f001:**
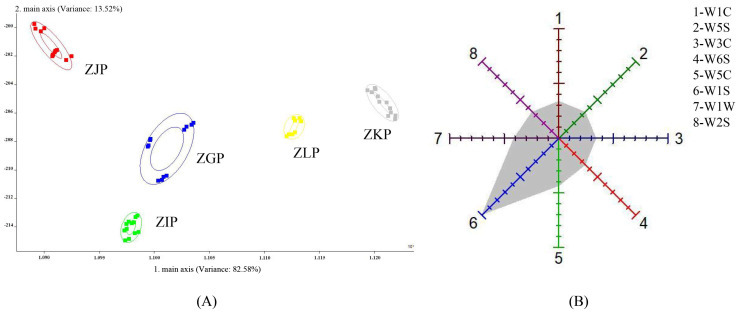
Linear discriminant analysis (LDA) and radar chart of the E−nose response data (**A**), LDA; (**B**), Radar chart.

**Figure 2 molecules-28-05326-f002:**
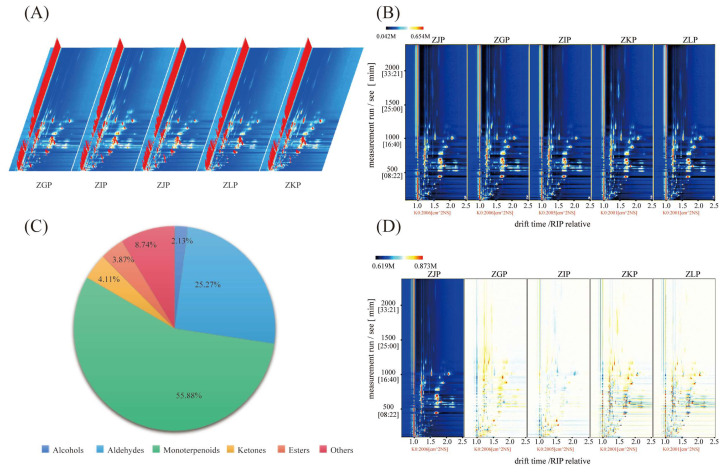
The CRZP at different stages of maturity: (**A**) Three−dimensional topographic plots; (**B**) Circular graph of relative content of various volatile organic compounds (VOCs); (**C**) Two−dimensional topographic plots; (**D**) Difference comparison topographic plots.

**Figure 3 molecules-28-05326-f003:**
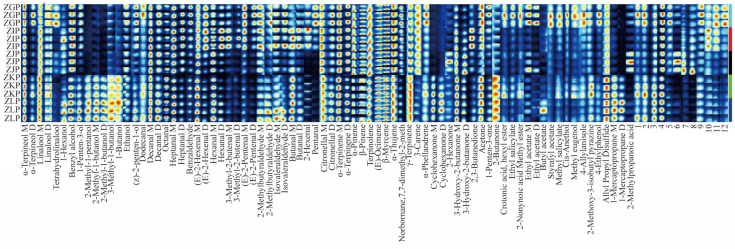
The VOC fingerprint of CRZP at different picking stages.

**Figure 4 molecules-28-05326-f004:**
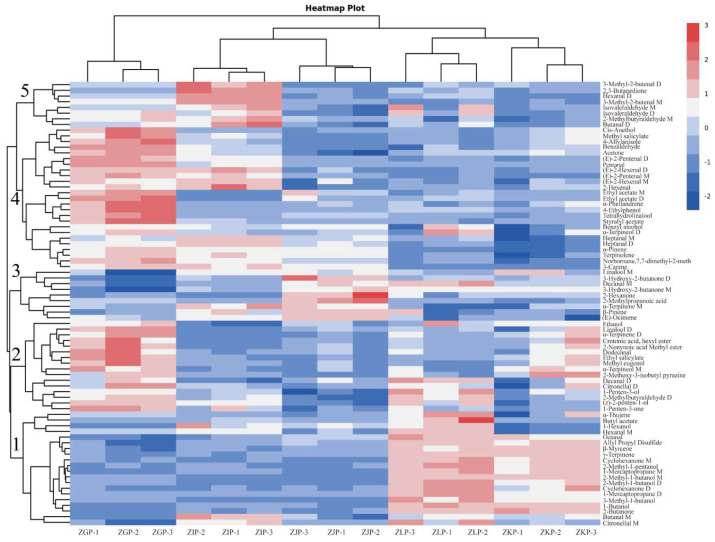
The Cluster heat map of VOC compounds in the CRZP at different harvesting periods. Each row represents a compound, and each column represents a group. The peaks of each compound are shown in different colours, with darker reds representing larger peaks and darker blues representing smaller peaks.

**Figure 5 molecules-28-05326-f005:**
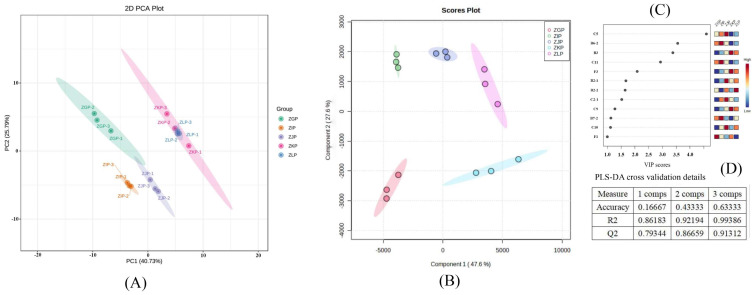
The principal component analysis (PCA) and partial least squares-discriminant analysis (PLS−DA) are based on the headspace−gas chromatography−ion mobility spectrometry (HS−GC−IMS) data. The figure includes (**A**) a 2D PCA plot; (**B**) PLS−DA score plot; (**C**) VIP scores; and (**D**) cross−validation results.

**Figure 6 molecules-28-05326-f006:**
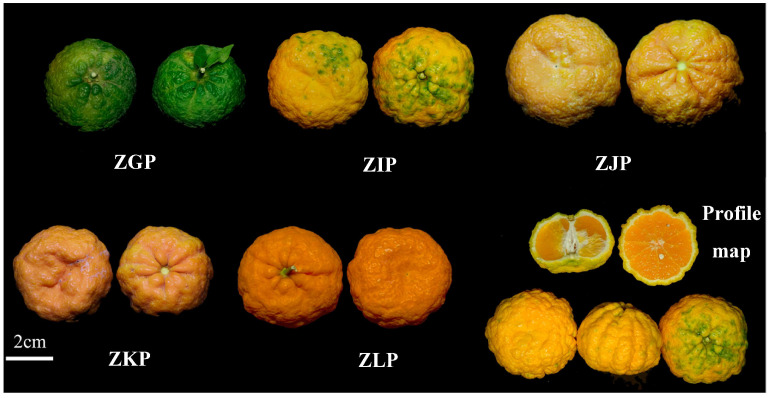
*Citrus reticulata* cv. Manau Gan (Zangju, CRZP) samples from each stage.

**Table 1 molecules-28-05326-t001:** Volatile organic compounds were identified in dry peel samples of Zangju at different ripening stages by HS-GC–IMS.

Count	Compounds	CAS#	Formula	MW	RI ^a^	Rt ^b^	Dt ^c^	Identification Approach
			Alcohols					
A1	1-Hexanol	111-27-3	C_6_H_14_O	102.2	876.9	356.1	1.3275	RI, Dt
A2	Benzyl alcohol	100-51-6	C_7_H_8_O	108.1	1026.5	618.156	1.1718	RI, Dt
A3	1-Penten-3-ol	616-25-1	C_5_H_10_O	86.1	689.1	176.263	0.9482	RI, Dt
A4	2-Methyl-1-pentanol	105-30-6	C_6_H_14_O	102.2	825.6	294.843	1.2882	RI, Dt
A5	2-Methyl-1-butanol M ^d^	137-32-6	C_5_H_12_O	88.1	739.9	213.899	1.2352	RI, Dt
A6	2-Methyl-1-butanol D ^e^	137-32-6	C_5_H_12_O	88.1	738.8	212.957	1.4757	RI, Dt
A7	3-Methyl-1-butanol	123-51-3	C_5_H_12_O	88.1	732.9	208.251	1.2467	RI, Dt
A8	1-Butanol	71-36-3	C_4_H_10_O	74.1	670.4	167.432	1.1837	RI, Dt
A9	Ethanol	64-17-5	C_2_H_6_O	46.1	495.1	104.055	1.1338	RI, Dt
A10	(z)-2-penten-1-ol	1576-95-0	C_5_H_10_O	86.1	774.8	244.291	0.9459	RI, Dt
			Aldehydes					
B1	Dodecanal	112-54-9	C_12_H_24_O	184.3	1441.7	2148.465	1.6582	RI, Dt
B2-1	Decanal M	112-31-2	C_10_H_20_O	156.3	1189.3	1007.691	1.5362	RI, Dt
B2-2	Decanal D	112-31-2	C_10_H_20_O	156.3	1187.3	1001.477	2.0588	RI, Dt
B3	Octanal	124-13-0	C_8_H_16_O	128.2	1012.7	593.035	1.8277	RI, Dt
B4-1	Heptanal M	111-71-7	C_7_H_14_O	114.2	906.5	398.247	1.3308	RI, Dt
B4-2	Heptanal D	111-71-7	C_7_H_14_O	114.2	903.2	393.121	1.7000	RI, Dt
B5	Benzaldehyde	100-52-7	C_7_H_6_O	106.1	966.1	501.991	1.1542	RI, Dt
B6-1	(E)-2-Hexenal M	6728-26-3	C_6_H_10_O	98.1	860.6	335.357	1.1867	RI, Dt
B6-2	(E)-2-Hexenal D	6728-26-3	C_6_H_10_O	98.1	854.8	328.236	1.5249	RI, Dt
B7-1	Hexanal M	66-25-1	C_6_H_12_O	100.2	797.5	265.933	1.2560	RI, Dt
B7-2	Hexanal D	66-25-1	C_6_H_12_O	100.2	798.4	266.823	1.5683	RI, Dt
B8-1	3-Methyl-2-butenal M	107-86-8	C_5_H_8_O	84.1	786.4	255.252	1.0934	RI, Dt
B8-2	3-Methyl-2-butenal D	107-86-8	C_5_H_8_O	84.1	786.5	255.335	1.3647	RI, Dt
B9-1	(E)-2-Pentenal M	1576-87-0	C_5_H_8_O	84.1	756.5	227.877	1.1106	RI, Dt
B9-2	(E)-2-Pentenal D	1576-87-0	C_5_H_8_O	84.1	756.5	227.877	1.3650	RI, Dt
B10-1	2-Methylbutyraldehyde M	96-17-3	C_5_H_10_O	86.1	670.6	167.536	1.1586	RI, Dt
B10-2	2-Methylbutyraldehyde D	96-17-3	C_5_H_10_O	86.1	670.8	167.61	1.4006	RI, Dt
B11-1	Isovaleraldehyde M	590-86-3	C_5_H_10_O	86.1	657.5	161.68	1.1744	RI, Dt
B11-2	Isovaleraldehyde D	590-86-3	C_5_H_10_O	86.1	658.7	162.188	1.4138	RI, Dt
B12-1	Butanal M	123-72-8	C_4_H_8_O	72.1	559.5	123.921	1.1102	RI, Dt
B12-2	Butanal D	123-72-8	C_4_H_8_O	72.1	560.3	124.194	1.2772	RI, Dt
B13	Cis-2-Hexenal	505-57-7	C_6_H_10_O	98.1	846.1	317.917	1.5202	RI, Dt
B14	Pentanal	110-62-3	C_5_H_10_O	86.1	704.2	186.665	1.4317	RI, Dt
			Monoterpenoids					
C1-1	α-Terpineol M	98-55-5	C_10_H_18_O	154.3	1170.8	953.005	1.2955	RI, Dt
C1-2	α-Terpineol D	98-55-5	C_10_H_18_O	154.3	1169.6	949.706	1.7928	RI, Dt
C2-1	Linalool M	78-70-6	C_10_H_18_O	154.3	1104.5	781.237	1.2254	RI, Dt
C2-2	Linalool D	78-70-6	C_10_H_18_O	154.3	1101.8	774.947	1.7674	RI, Dt
C3	Tetrahydrolinalool	78-69-3	C_10_H_22_O	158.3	1135.6	857.495	1.2787	RI, Dt
C4-1	α-Terpinene M	99-86-5	C_10_H_16_	136.2	1026.9	619	1.2185	RI, Dt
C4-2	α-Terpinene D	99-86-5	C_10_H_16_	136.2	1027.5	620.072	1.7329	RI, Dt
C5	α-Pinene	80-56-8	C_10_H_16_	136.2	939.9	453.351	1.7396	RI, Dt
C6	β-Pinene	127-91-3	C_10_H_16_	136.2	978.5	526.824	1.6425	RI, Dt
C7	Terpinolene	586-62-9	C_10_H_16_	136.2	1077.3	719.981	1.2254	RI, Dt
C8	(E)-Ocimene	3779-61-1	C_10_H_16_	136.2	1045.9	655.274	1.7067	RI, Dt
C9	β-Myrcene	123-35-3	C_10_H_16_	136.2	995.7	563.133	1.7204	RI, Dt
C10	α-Thujene	2867-05-2	C_10_H_16_	136.2	928.5	433.702	1.6680	RI, Dt
C11	Norbornane,7,7-dimethyl-2-meth	471-84-1	C_10_H_16_	136.2	953.1	477.273	1.2235	RI, Dt
C11	α-Phellandrene	99-83-2	C_10_H_16_	136.2	1155	909.13	1.2225	RI, Dt
C12	γ-Terpinene	99-85-4	C_10_H_16_	136.2	1058.5	680.426	1.7029	RI, Dt
C13	3-Carene	13466-78-9	C_10_H_16_	136.2	1015.2	597.656	1.2188	RI, Dt
C14-1	Citronellal M	106-23-0	C_10_H_18_O	154.3	1144.3	880.222	1.3530	RI, Dt
C14-2	Citronellal D	106-23-0	C_10_H_18_O	154.3	1142.3	875.113	1.9004	RI, Dt
			Ketones					
D1-1	Cyclohexanone M	108-94-1	C_6_H_10_O	98.1	896.2	382.595	1.1568	RI, Dt
D1-2	Cyclohexanone D	108-94-1	C_6_H_10_O	98.1	895.1	380.964	1.4568	RI, Dt
D2	2-Hexanone	591-78-6	C_6_H_12_O	100.2	789.8	258.447	1.1900	RI, Dt
D3-1	3-Hydroxy-2-butanone M	513-86-0	C_4_H_8_O_2_	88.1	720.6	198.749	1.0570	RI, Dt
D3-2	3-Hydroxy-2-butanone D	513-86-0	C_4_H_8_O_2_	88.1	719.4	197.846	1.3359	RI, Dt
D4	2,3-Butanedione	431-03-8	C_4_H_6_O_2_	86.1	585	132.781	1.1734	RI, Dt
D5	Acetone	67-64-1	C_3_H_6_O	58.1	518.5	110.878	1.1178	RI, Dt
D6	1-Penten-3-one	1629-58-9	C_5_H_8_O	84.1	692.3	178.449	1.0764	RI, Dt
D7	2-Butanone	78-93-3	C_4_H_8_O	72.1	594.3	136.207	1.0606	RI, Dt
			Esters					
E1	Crotonic acid, hexyl ester	19089-92-0	C_10_H_18_O_2_	170.3	1374.5	1756.51	1.4579	RI, Dt
E2	Ethyl salicylate	118-61-6	C_9_H_10_O_3_	166.2	1283.7	1337.581	1.2676	RI, Dt
E3	2-Nonynoic acid Methyl ester	111-80-8	C_10_H_16_O_2_	168.2	1352.3	1642.908	1.4606	RI, Dt
E4-1	Ethyl acetate M	141-78-6	C_4_H_8_O_2_	88.1	613.2	143.383	1.0996	RI, Dt
E4-2	Ethyl acetate D	141-78-6	C_4_H_8_O_2_	88.1	611.6	142.749	1.3408	RI, Dt
E5	Butyl acetate	123-86-4	C_6_H_12_O_2_	116.2	810	278.372	1.2387	RI, Dt
E6	Styralyl acetate	93-92-5	C_10_H_12_O_2_	164.2	1220.4	1106.023	1.0544	RI, Dt
E7	Methyl salicylate	119-36-8	C_8_H_8_O_3_	152.1	1200.2	1040.893	1.2011	RI, Dt
			Others					
F1	Cis-Anethol	104-46-1	C_10_H_12_O	148.2	1371.5	1740.328	1.2241	RI, Dt
F2	Methyl eugenol	93-15-2	C_11_H_14_O_2_	178.2	1404	1919.024	1.4437	RI, Dt
F3	4-Allylanisole	140-67-0	C_10_H_12_O	148.2	1239.3	1170.509	1.2348	RI, Dt
F4	2-Methoxy-3-isobutyl pyrazine	24683-00-9	C_9_H_14_N_2_O	166.2	1212.8	1081.021	1.2998	RI, Dt
F5	4-Ethylphenol	123-07-9	C_8_H_10_O	122.2	1157.2	914.964	1.1964	RI, Dt
F6	Allyl Propyl Disulfide	2179-59-1	C_6_H_12_S_2_	148.3	1068.8	701.854	1.7138	RI, Dt
F7-1	1-Mercaptopropane M	107-03-9	C_3_H_8_S	76.2	626.5	148.635	1.1690	RI, Dt
F7-2	1-Mercaptopropane D	107-03-9	C_3_H_8_S	76.2	624.1	147.654	1.3602	RI, Dt
F8	2-Methylpropanoic acid	79-31-2	C_4_H_8_O_2_	88.1	780.7	249.895	1.1613	RI, Dt
			unidentified					
U1	1	unidentified	*	0	1289.5	1360.959	1.6034	RI, Dt
U2	2	unidentified	*	0	1240	1172.995	1.3670	RI, Dt
U3	3	unidentified	*	0	1124.8	830.153	1.1964	RI, Dt
U4	4	unidentified	*	0	1087	741.255	1.1907	RI, Dt
U5	5	unidentified	*	0	908.3	401.022	1.2154	RI, Dt
U6	6	unidentified	*	0	782.6	251.715	1.1745	RI, Dt
U7	7	unidentified	*	0	695.9	180.852	1.1052	RI, Dt
U8	8	unidentified	*	0	691.9	178.146	1.3284	RI, Dt
U9	9	unidentified	*	0	586.8	133.443	1.0226	RI, Dt
U10	10	unidentified	*	0	598.2	137.635	1.1155	RI, Dt
U11	11	unidentified	*	0	597.9	137.522	1.2236	RI, Dt
U12	12	unidentified	*	0	563	125.098	1.2115	RI, Dt

Note: MW: molecular mass. ^a^ Represents the retention index (RI) calculated using n-ketones C4-9 as an external standard. ^b^ Represents the retention time (Rt) in the capillary GC column. ^c^ Represents the drift time (Dt) in the drift tube. ^d^ Represents the monomer (M) of the substance. ^e^ Represents the dimer (D) of the substance. * Represents an undefined substance.

**Table 2 molecules-28-05326-t002:** Relative content of VOCs in the CRZP at different harvesting periods.

Sample	Alcohols	Aldehydes	Monoterpenoids	Ketones	Esters	Others	Total
	Peak Intensity						
ZGP	4057.40 ± 85.36 2.13 ± 0.0531%	48,216.08 ± 669.70 25.27 ± 0.3588%	106,640.88 ± 1889.00 55.88 ± 0.6799%	7848.07 ± 414.90 4.12 ± 0.3356%	7385.46 ± 663.58 3.87 ± 0.3571%	16,678.18 ± 652.85 8.74 ± 0.2549%	190,826.08 ± 2979.08 100%
ZIP	3503.93 ± 223.57 1.99 ± 0.1516%	48,588.76 ± 436.03 27.62 ± 0.3455%	97,652.15 ± 136.62 55.50 ± 0.1949%	9587.01 ± 508.14 5.45 ± 0.5158%	3536.75 ± 78.47 2.01 ± 0.0462%	13,072.33 ± 143.86 7.43 ± 0.0735%	175,940.95 ± 606.68 100%
ZJP	3316.51 ± 174.06 1.98 ± 0.0819%	40,396.04 ± 2093.32 24.15 ± 0.8953%	96,535.55 ± 917.79 57.77 ± 0.9334%	10,376.60 ± 654.61 6.21 ± 0.5158%	3662.04 ± 355.46 2.19 ± 0.2065%	12,854.95 ± 287.34 7.69 ± 0.1105%	167,141.68 ± 3621.46 100%
ZKP	4553.94 ± 241.48 2.78 ± 0.1011%	39,014.17 ± 2260.32 23.83 ± 0.4442%	89,086.27 ± 6168.29 54.37 ± 0.3045%	9727.70 ± 268.59 5.95 ± 0.3045%	5098.73 ± 914.90 3.09 ± 0.4344%	16,364.86 ± 1310.06 9.98 ± 0.1752%	163,845.67 ± 11,056.22 100%
ZLP	5734.82 ± 316.98 3.33 ± 0.2417%	45,457.91 ± 926.48 26.37 ± 0.4901%	92,519.90 ± 172.36 53.67 ± 0.3186%	9727.17 ± 285.91 5.64 ± 0.2818%	3918.24 ± 189.80 2.27 ± 0.1268%	15,027.25 ± 412.60 8.72 ± 0.2762%	172,385.29 ± 911.82 100%

Note: VOCs: Volatile Organic Compounds. CRZP: Zangju (*Citrus reticulata* cv. Manau Gan) peel. 3.6 PCA and PLS-DA analysis of VOC-based fingerprints.

## Data Availability

Data are contained within the article/Appendix A.

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
