# Peer review of "Electronic Nose and Head Space GC–IMS Provide Insights into the Dynamic Changes and Regularity of Volatile Compounds in Zangju (Citrus reticulata cv. Manau Gan) Peel at Different Maturation Stages"

_molecules, 2023, doi:10.3390/molecules28145326_

Round 1
Reviewer 1 Report
The study of changes in volatile components provides insight into the evolution of metabolites at different maturation states. Is Table 1 a list of compounds identified at all maturation stages? There should be a more comprehensive caption.
In Table 2 I would not put the intensity of the peaks (because it is a repetition of the material in supplementary), but the amounts (% or mg) of each class of compound.
Why do VOCs decrease in December and January?
There is an error in Table 2: replace ZJP with ZKP indicating January.
I would improve the quality of the image figure 1.
Figure 3 is not very clear and needs a more detailed caption.
Author Response
Dear reviewer,
Thanks for providing us with this great opportunity to submit a revised version of our manuscript. We appreciate the detailed and constructive comments provided by the editor, reviewer 1 and 2. We have carefully revised the manuscript by incorporating all the suggestions by the review panel.
We hope this revised manuscript has addressed your concerns, and look forward to hearing from you.
Sincerely,
Youping Liu.
Comment 1:
The study of changes in volatile components provides insight into the evolution of metabolites at different maturation states. Is Table 1 a list of compounds identified at all maturation stages? There should be a more comprehensive caption.
Response 1:
Thank you for the detailed review. Table 1 lists the volatile components of all mature stages, which are described in section 2.3 of the article. Due to technical issues, there are still 12 volatile compounds that cannot be identified, which will be our follow-up research direction.
Comment 2:
In Table 2 I would not put the intensity of the peaks (because it is a repetition of the material in supplementary), but the amounts (% or mg) of each class of compound.
Response 2:
Thank you for the detailed review. We have carefully considered your suggestion and listed the relative content of various compounds in percentage form in Table 2, represented in red font.
Comment 3:
Why do VOCs decrease in December and January?
Response 3:
Thanks for your great suggestion on improving the accessibility of our manuscript. We have carefully considered your suggestion and modifications to the first paragraph of section 2.5 in the article, marked in red (Lines 316-318).
Comment 4:
There is an error in Table 2: replace ZJP with ZKP indicating January.
Response 4:
Thank you for the detailed review. We have made revisions in Table 2 of the article.
Comment 5:
I would improve the quality of the image figure 1.
Response 5:
Thank you for the detailed review. We have revised the quality of the Figure 1 and included high-quality Figure 1 in the re-manuscript.
Comment 6:
Figure 3 is not very clear and needs a more detailed caption.
Response 6:
Thank you for the detailed review. We have carefully considered your suggestion and revised section 2.2 in the article (Lines 129-155).
Reviewer 2 Report
The manuscript entitled “ E-nose and HS-GC-IMS provide insights into the dynamic changes and regularity of volatile components in Zangju (Citrus reticulata cv. Manau Gan) peel at different maturation stages” by Peng Wang et al., reports on the evaluation of the volatile compounds of dried Zangju peel, and analyze the role of dynamic changes at different maturation stages using electronic nose, headspace-gas chromatography-ion mobility spectrometry (HS-GC-IMS) and partial least squares-discriminant analysis (PLS-DA) methods.
The theme of the manuscript is of scientific interest. However, before publication, some corrections and improvements are needed.
Firstly the manuscript must be submitted in the Molecules journal template available on the Instructions for Authors page at https://www.mdpi.com/journal/molecules/instructions
TITLE
Use “ ... Volatile Compounds ...” instead “ ... VOlatile Components ...”
ABSTRACT
Please update the sentence regarding the most dominant chemical families as follow : “ ... 78 volatile organic compounds (VOMs), belonging to different chemical families mainly monoterpenes (33.35%), aldehydes (29.61%) and alcohols (20.31%), were found ...”
Please update the following statement “ ... which could be used as flavour markers of CRZP collected samples ... “ because it is not correct. We cannot state that the 12 VIPs are flavour markers of CRZP samples. The 12 VIPs allow the separating/discriminate of the CRZP samples among the different maturity stages. However, to be flavour markers of CRZP, they should be identified only in these kinds of samples. However, the identified VIPs are also identified in other kinds of fruits and peels, and food matrices. So the adequate form is “ ... which could be used to discriminate CRZP samples at the investigated maturity stages ... “
INTRODUCTION
“ ... mainly includes mandarin (Citrus reticulata), grapefruit (Citrus maxima) and citron (Citrus medica). ...” and the oranges and limes???? Are missing and must be considered.
Please change as follow “ ... Zangju (C. reticulata cv. Manau Gan), among others. Zangju is known locally ... “
“ ... including mainly terpenoids, esters, alcohols, aldehydes, among others [7, 8], have antioxidant, anti-inflammatory, ...”
“ ... used mainly in cooking, food additives, and cosmetic additives. The dried peel is traditional ... “
“ ... hot air drying and freeze drying, and different drying methods which influence the volatile composition of citrus peel. ...”
Please use all over the manuscript the term “volatile compounds” instead “volatile components”
“ ... volatile organic compounds (VOCs) in the drying process ... “ the abbreviation “VOCs” is already described in the abstract, so here use only “ ... VOCs in the drying process ... “ instead “ ... volatile organic compounds (VOCs) in the drying process ... “
Do you use LDA and PLS-DA? the both?? Why???
C. reticulata varieties, C. reticulata must be in italic mode
RESULTS
Figure 2B instead Fig. 2B because is the beginning of the sentence
Change “ ... induction of volatile components in the samples . “ by “ ... induction of VOCs in the samples ... “
“ ... such as alcohols, aldehydes, esters, etc. ...” by “ ... such as alcohols, aldehydes, and esters, ...”
Please update Table 1. The meaning of the upper-case letters M and D must be described. The same for MW, RI, Rt, Dt.
The footnotes of Table 1 are not indicated in the table.
The first 5 alcohols described in Table 1 must be included in the Monoterpenoids group and not in the alcohols chemical family. They are C10 compounds.
The compound B13 2-Hexenal must be specified the isómer, cis- or trans-???
The axes of all figures need to be improved since we cannot read anything
Table 2 The meaning of the samples abbreviations must be described as table footnotes
Author Response
Dear reviewer,
Thanks for providing us with this great opportunity to submit a revised version of our manuscript. We appreciate the detailed and constructive comments provided by the editor, reviewer 1 and 2. We have carefully revised the manuscript by incorporating all the suggestions by the review panel.
We hope this revised manuscript has addressed your concerns, and look forward to hearing from you.
Sincerely,
Youping Liu.
Comment 1:
Firstly the manuscript must be submitted in the Molecules journal template available on the Instructions for Authors page at https://www.mdpi.com/journal/molecules/instructions.
Response 1:
Thank you for the detailed review. We have revised according to the template of Molecules journal, please refer to the re-manuscript for details.
Comment 2:
TITLE. Use “ ... Volatile Compounds ...” instead “ ... VOlatile Components ...”.
Response 2:
Thank you for the detailed review. We have made modifications in the revised manuscript and marked it in blue (Line 3).
Comment 3:
INTRODUCTION. “... mainly includes mandarin (Citrus reticulata), grapefruit (Citrus maxima) and citron (Citrus medica). ...” and the oranges and limes???? Are missing and must be considered.
Response 3:
Thank you for the detailed review. We have carefully considered your suggestion and added oranges and limes to the INTRODUCTION section, marked in blue (Line 33).
Comment 4:
INTRODUCTION. Please change as follow “... Zangju (C. reticulata cv. Manau Gan), among others. Zangju is known locally ... ”.
“... including mainly terpenoids, esters, alcohols, aldehydes, among others [7, 8], have antioxidant, anti-inflammatory, ...”
“... used mainly in cooking, food additives, and cosmetic additives. The dried peel is traditional ...”
“... hot air drying and freeze drying, and different drying methods which influence the volatile composition of citrus peel. ...”
Response 4:
Thank you for the detailed review. Based on your suggestion, we have made revisions in the relevant parts of the article and marked them in blue (Lines 35, 36, 54 ,55, 58, 59, 61-63).
Comment 5:
INTRODUCTION. Please use all over the manuscript the term “volatile compounds” instead “volatile components”.
“... volatile organic compounds (VOCs) in the drying process ...” the abbreviation “VOCs” is already described in the abstract, so here use only “... VOCs in the drying process ...” instead “... volatile organic compounds (VOCs) in the drying process ...”.
Response 5:
Thank you for the detailed review. We have carefully considered your suggestion and made modifications in the relevant parts of the article, marked in blue (Line 66).
Comment 6:
INTRODUCTION. Do you use LDA and PLS-DA? the both?? Why???
Response 6:
Thank you for the detailed review. Based on our understanding and previous research, we chose two models because LDA model can better characterize the differences between groups, so that we can decide to adopt more accurate technology (HS-GC-IMS) to determine the volatile components in CRZP after Electronic nose measurement. The PLD-DA model is designed to better understand the key components that change during the maturation process of CRZP.
Comment 7:
INTRODUCTION. C. reticulata varieties, C. reticulata must be in italic mode.
Response 7:
Thank you for the detailed review. We have made modifications in the article and marked it in blue.
Comment 8:
RESULTS. Figure 2B instead Fig. 2B because is the beginning of the sentence.
Response 8:
Thank you for the detailed review. We have made modifications in the article and marked it in blue (Line 111).
Comment 9:
RESULTS. Change “... induction of volatile components in the samples.” by “... induction of VOCs in the samples ...”.
“... such as alcohols, aldehydes, esters, etc. ...” by “... such as alcohols, aldehydes, and esters, ...”.
Response 9:
Thank you for the detailed review. We have made modifications in section 2.1 of the article and marked it in blue (Lines 112-115).
Comment 10:
RESULTS. Please update Table 1. The meaning of the upper-case letters M and D must be described. The same for MW, RI, Rt, Dt.
The footnotes of Table 1 are not indicated in the table.
Response 10:
Thank you for the detailed review. We have carefully considered your suggestion and provided additional explanations in Table 1 (Lines 257-260).
Comment 11:
RESULTS. The first 5 alcohols described in Table 1 must be included in the Monoterpenoids group and not in the alcohols chemical family. They are C10 compounds.
Response 11:
Thanks for your great suggestion on improving the accessibility of our manuscript. We have carefully considered your suggestion and made revisions. The revised Tables and Figures have been presented in the re-manuscript.
Comment 12:
RESULTS. The compound B13 2-Hexenal must be specified the isómer, cis- or trans-???
Response 12:
Thank you for the detailed review. We carefully checked the results of 2-Hexenal and determined that it is Cis-2-Hexenal.
Comment 13:
RESULTS. The axes of all figures need to be improved since we cannot read anything.
Response 13:
Thanks for your great suggestion on improving the accessibility of our manuscript.We have improved the quality of the images to make the coordinate axes clearer, and have made revisions in the article.
Comment 14:
RESULTS. Table 2 The meaning of the samples abbreviations must be described as table footnotes.
Response 14:
Thank you for the detailed review. We annotated the abbreviations in the form of footnotes in Table 2 and marked them in blue (Line 364).
Round 2
Reviewer 2 Report
The authors follow and consider almost all the suggestions made in the previous revision. However, some minor typos need to be updated.
Abstract
“ ... comprehensively evaluate the volatile compounds of dried CRZP ... “
“ .. The PLS-DA model results showed that the samples collected at different maturity stages could be effectively discriminated ...”
In the sentence “ ... Specifically, the relative VOCs content of CRZP ... “ the abbreviation VOCs appears without meaning. Therefore I suggest “ ... Specifically, the relative volatile organic compounds (VOCs) content of CRZP ... “
RESULTS
The resolution of Figure 2 must be improved since we cannot read the letters /numbers. the same for Figure 3
The caption of Table 1 “ ... samples by HS-GC-IMS.” correct
The compounds B15-1Citronellal M and B15-2 Citronellal D are monoterpenoids and therefore must be included in the proper chemical group.
Please correct the numerical values in Table 2 (4,057.40 ± 85.36). What means the comma? Must be 4057.40, and so on.
Author Response
Dear reviewer,
Thanks for providing us with this great opportunity to submit a revised version of our manuscript. We appreciate the detailed and constructive comments provided by you. We have carefully revised the manuscript by incorporating all the suggestions by the review panel.
We hope this revised manuscript has addressed your concerns, and look forward to hearing from you.
Sincerely,
Youping Liu.
Responses to the comments from you.
Comment 1:
Abstract. “... comprehensively evaluate the volatile compounds of dried CRZP ...”.
“.. The PLS-DA model results showed that the samples collected at different maturity stages could be effectively discriminated ...”.
In the sentence “... Specifically, the relative VOCs content of CRZP ... “ the abbreviation VOCs appears without meaning. Therefore I suggest “ ... Specifically, the relative volatile organic compounds (VOCs) content of CRZP ...”.
Response 1:
Thank you for the detailed review. Based on your suggestion, we have made revisions in the relevant parts of the article and marked them in blue (Lines 13, 14, 19, 20, 22, 23).
Comment 2:
RESULTS. The resolution of Figure 2 must be improved since we cannot read the letters /numbers. the same for Figure 3.
Response 2:
Thanks for your great suggestion on improving the accessibility of our manuscript. We have improved the quality of the images to make the coordinate axes clearer, and have made revisions in the article (Lines 156-160, 290-292).
Comment 3:
RESULTS. The caption of Table 1 “... samples by HS-GC-IMS.” correct.
Response 3:
Thank you for the detailed review. We have corrected the title of Table 1 (Line 249).
Comment 4:
RESULTS. The compounds B15-1Citronellal M and B15-2 Citronellal D are monoterpenoids and therefore must be included in the proper chemical group.
Response 4:
Thanks for your great suggestion on improving the accessibility of our manuscript. We have carefully considered your suggestion and made revisions. The revised Tables and Figures have been presented in the re-manuscript.
Comment 5:
RESULTS. Please correct the numerical values in Table 2 (4,057.40 ± 85.36). What means the comma? Must be 4057.40, and so on.
Response 5:
Thank you for the detailed review. We have made modifications to the data in Table 2 based on your suggestion and marked it in blue.